# Mechanical Properties and Interfacial Characterization of Additive-Manufactured CuZrCr/CoCrMo Multi-Metals Fabricated by Powder Bed Fusion Using Pulsed Wave Laser

**DOI:** 10.3390/mi15060765

**Published:** 2024-06-07

**Authors:** Hao Zhang, Xiang Jin, Zhongmin Xiao, Liming Yao

**Affiliations:** 1School of Mechanical and Aerospace Engineering, Nanyang Technological University, 50 Nanyang Avenue, Singapore 639798, Singapore; zhanghaob@sjtu.edu.cn; 2Institute of Materials Modification and Modelling, School of Materials Science and Engineering, Shanghai Jiao Tong University, Shanghai 200240, China; 3School of Mechanical Science and Engineering, Northeast Petroleum University, Daqing 163318, China; 13653458637@163.com; 4Zhengzhou Research Institute, Harbin Institute of Technology, Zhengzhou 450002, China

**Keywords:** additive manufacturing, dissimilar alloy, interface strength, pulsed wave laser

## Abstract

In this study, CoCrMo cuboid samples were deposited on a CuZrCr substrate using laser powder bed fusion (L-PBF) technology to investigate the influence of process parameters and laser remelting strategies on the mechanical properties and interface characteristics of multi-metals. This study found that process parameters and laser scanning strategies had a significant influence on the mechanical properties and interface characteristics. Samples fabricated with an $E_{V}$ ≤ 20 J/mm^3^ showed little tensile ductility. As the volumetric energy density ($E_{V}$) increased to a range between 40 J/mm^3^ and 100 J/mm^3^, the samples achieved the desired mechanical properties, with a strong interface combining the alloys. However, an excessive energy density could result in cracks due to thermal stress. Laser remelting significantly improved the interface properties, especially when the $E_{V}$ was below 40 J/mm^3^. Variances in the $E_{V}$ showed little influence on the hardness at the CuZrCr end, while the hardness at the interface and the CoCrMo end showed an increasing and decreasing trend with an increase in the $E_{V}$, respectively. Interface characterization showed that when the $E_{V}$ was greater than 43 J/mm^3^, the main defects in the L-PBF CoCrMo samples were thermal cracks, which gradually changed to pores with a lack of fusion when the $E_{V}$ decreased. This study provides theoretical and technical support for the manufacturing of multi-metal parts using L-PBF technology.

## 1. Introduction

In recent years, additive manufacturing (AM) has emerged as a revolutionary technology, transforming traditional manufacturing processes across industries. Unlike conventional manufacturing methods, AM allows for the precise deposition of multiple materials, enabling the direct fabrication of complex structures with unique properties [1].

However, there has been a lack of focus on using the L-PBF technique to manufacture multi-metal parts [2]. Utilizing multi-metals in a single component can integrate the advantages of each material while avoiding their deficiencies. Combining copper alloys and cobalt alloys can achieve a synergistic enhancement in material properties, leveraging the excellent thermal conductivity of copper alloys with the high-temperature mechanical properties and corrosion resistance of cobalt alloys. This unique combination of properties makes the dissimilar alloy highly suitable for a wide range of advanced applications. For example, in the aerospace industry, this dissimilar alloy can be used in the construction of engine components and other critical parts that must withstand extreme temperatures and corrosive environments. In the nuclear industry, the superior thermal conductivity and corrosion resistance are crucial for the reliability and safety of nuclear reactors. Additionally, Cu/Co multi-alloy parts can be used in thermal management systems where both high thermal conductivity and mechanical robustness are essential [3,4,5].

Multi-metal parts were conventionally fabricated by joining processes such as welding [6,7,8,9]. However, these joining technologies were mainly applied when fabricating large components with a multi-metal interface and simple geometry. When encountering small-size parts or intricate components with considerably more complex interface shapes, welding technology may have limited efficacy. For example, a rocket engine thrust chamber featuring complex internal flow passages requires an inner part of high thermal conductivity Cu and an outer part of an Ni-based alloy with high-temperature strength, which can be only manufactured by AM technology.

Despite its wide range of industrial applications, there is still inadequate research on the AM of multi-metals. One of the key issues is creating strong metallurgical bonds between different metals because of variations in their thermal, physical, and chemical properties. Weak metallurgical bonds can result in weak interfaces and deteriorating mechanical properties and reliability. This problem can be exacerbated by the high thermal stress induced by the rapid melting and solidification intrinsic to the L-PBF process [10]. Additionally, incomplete solubility between different metals at the interface forms brittle intermetallic compounds or amorphous structures [11,12], which result in a decline in plasticity. Hendrik Hotz et al. [13] investigated the local chemical composition, microstructure, and mechanical properties in the interfacial region of Ti/Al structures manufactured using laser-based direct energy deposition (L-DED). Functionally graded structures with step transitions from an AlMg_3_ substrate to Ti6Al4V and graded transitions with AlSi10Mg transition layers were both manufactured and characterized. The results showed that macroscopic cracks were found in the graded transition samples, which developed in the AlSi10Mg–TiAl6V4 layer and extended across the entire width of the structure. This was attributed to the large amounts of intermetallic phases formed in the transition layers. For the graded transition samples, a layer of IMPs consisting of needle-like Ti_5_Si_3_ and two types of titanium aluminide were formed in the AlSi10Mg–Ti6Al4V transition region.

Designing manufacturing strategies [14,15,16,17], creating interlocked interface structures [18,19,20,21], conducting heat treatments in post-processing [22,23,24,25], and introducing transition layers [4,26,27,28] are the main methods that have been proposed to improve the interface properties in additive-manufactured multi-metals. Up to now, various manufacturing strategies such as process parameter optimization, laser remelting, substrate preheating, and developing functionally graded structures have been proposed to be applied to improve the interface properties. Li et al. [29] conducted a 673 K preheating of a substrate to manufacture Ti6Al4V/NiTi functionally graded samples using L-DED technology. Crack-free samples were successfully fabricated and exceptional tensile properties with a 202.4 MPa ultimate tensile strength and a fracture elongation of 6.71% were achieved. Liu et al. [16] investigated functional graded structures on the interfacial bonding characteristics of CuSn10/316 SS multi-metals. The results showed that a compositional gradient between the two materials could lead to a smooth transition of both the microstructure and hardness at the interface, relieving the stress concentration and reducing crack occurrence. However, excess compositional gradient layers could potentially lead to the penetration of Cu into Fe grains, which could induce Cu-penetration cracks in the interdiffusion zone.

In this study, CoCrMo and CuZrCr were selected to investigate the impact of a remelting strategy on the interface morphology and mechanical properties of additive-manufactured multi-metals. The influence of process parameters on the interface characteristics and mechanical properties of the CuZrCr/CoCrMo multi-metal samples were investigated in detail, providing a reference and guidance for the development and application of multi-material parts manufactured using L-PBF technology.

## 2. Materials and Methods

### 2.1. L-PBF Process Parameters

In this study, a Renishaw AM400 L-PBF device (Renishaw, Tokyo, Japan) was used to manufacture CoCrMo blocks on the CuZrCr substrate. The specific experimental steps were as follows: A CuZrCr substrate with dimensions of 120 mm × 120 mm × 8 mm was first placed on the building platform of the L-PBF machine and then CoCrMo metal powder with a size range of 10–45 µm was deposited on the substrate and flattened by a roller. A laser was then focused on the powder layer to melt the CoCrMo powder to a desired shape. Subsequently, a new layer of CoCrMo powder was deposited and flattened on the build-chamber platform. The new layer was again selectively melted by the laser. This process continued repeatedly until the required shape was obtained. A total of 53 CoCrMo cuboid samples with dimensions of 8 mm × 4 mm × 10 mm were manufactured on the CuZrCr substrate to investigate the influence of process parameters on the mechanical properties of the dissimilar alloys.

The AM400 used a fiber laser with a wavelength of 1070 nm and a laser diameter of 70 μm, which operated in a power-modulated PW mode. The working principle of the pulsed wave laser is illustrated in Figure 1. A total of 27 process parameter combinations were used in this experiment (Table 1). The scan velocity (V) was the ratio of the point distance ($D_{p}$) to the exposure time ($E_{t}$). The laser scan direction was rotated by 90° after each layer of powder was fabricated. In this study, the layer thickness (T) and the hatch spacing ($H_{s}$) were set to be 60 µm and 80 µm, respectively.

For pulsed laser transmitters, Equation (1) needs to be modified to Equation (2) to quantify the parameter combinations. Our results are shown in Table 1.
(1)$$E_{V}=P/\left(T\times H_{s}\times V\right)$$
(2)$$E_{V}=P\times \delta /\left(T\times H_{s}\times V\right)$$
where P denotes the laser power, $H_{s}$ is the hatch spacing, V is the scan velocity, $D_{p}$ is the point distance, and $E_{t}$ is the exposure time. The duty cycles (δ) were added to Equation (2) and ranged from 0.0 to 1.0, and served as a multiplier to account for the PW exposure parameter. According to a comparative study between continuous wave and pulse wave emissions in L-PBF [30], the duty cycles (δ) for exposure times ($E_{t}$) of 50 µs, 75 µs, and 100 µs are 0.54 s, 0.73 s, and 0.86 s, respectively. Comparatively, this would indicate that the δ for the $E_{t}$ selected in our print parameters of 50 µs, 80 µs, and 110 µs would be 0.54 s, 0.75 s, and 0.90 s, respectively, based on a simple factor of the ratio. The $E_{V}$ was calculated using Equation (2) and is listed in Table 1.

Two manufacturing strategies were investigated using the same process parameters during L-PBF manufacturing. The first manufacturing strategy did not involve laser remelting, while the second printing strategy involved laser remelting thrice within the fabrication of the first three layers.

### 2.2. Mechanical Properties: Tests

Mechanical tests were performed on a total of 54 samples fabricated using 27 combinations of different process parameters and two scanning strategies. Flat dog-bone samples for tensile tests with gauge section dimensions of 8 mm × 1 mm × 2 mm were extracted from the printed cuboid samples using wire cutting methods. To enhance the surface quality and eliminate visible surface cracks, all samples were polished using #220, #500, and #1200 SiC grit papers before the tensile test. The tensile properties of the samples were analyzed using a Shimadzu Autograph AG-X Plus 10kn (Shimadzu, Kyoto, Japan). A non-contact digital video extensometer (TRViewX, Shimadzu, Kyoto, Japan) was employed to measure the elongation. Fractures could either occur towards the CuZrCr or CoCrMo ends of the sample or at the interface, and the location for each specimen was recorded. Each sample was tested once.

Samples for the hardness test were mounted with phenolic resin using a Schneider Electric PRESSLAM 1.1. FutureTech (Taipei City, Taiwan). A FM-300e microhardness tester was used to analyze the hardness of the multi-metal samples with a specific load of 300 gf. All samples were successively ground using #220, #500, #1200, #2000, and #4000 SiC grit papers before the hardness test.

### 2.3. Interface Characterization

The samples for the interface characterization were mounted with phenolic resin using a Schneider Electric PRESSLAM 1.1. The mounted samples were then ground and polished according to standard procedures. Additionally, a non-dry OPU colloidal silica suspension (SiO_2_) was utilized for the final polishing of the samples before the microstructure characterization. The interface characterization was conducted using an Olympus LEXT OLS4100 (Olympus, Tokyo, Japan) 3D measuring laser microscope.

## 3. Results and Discussion

### 3.1. Tensile Properties

The optimization of processing parameters is significant to ensure the fabrication of multi-metal parts with desired properties. Tensile tests were conducted to investigate the influence of process parameters on the mechanical properties of the samples. The results of the tensile tests of all samples are recorded in Table 2. The strength and stress of the samples that suffered failures before the tensile test were recorded as 0.

Figure 2, Figure 3 and Figure 4 show the stress–strain curves of samples under different process parameters. Figure 5 shows the effect of the volumetric energy density on stress and strain.

In Figure 2, Figure 3 and Figure 4, (a) represents the samples with no laser remelting, while (b) represents the samples with the first three layers remelted thrice. By comparing (a) and (b) in Figure 2, Figure 3 and Figure 4, it can be observed that a particular pattern was followed by the stress–strain curves irrespective of the remelting strategies. The volumetric energy density had a significant influence on the tensile properties of the samples. Samples fabricated with an $E_{V}$ ≤ 20 J/mm^3^ suffered from poor metallurgical bonding at the interface. Most fractured under a considerably small strain. Tensile plasticity generally occurred in the samples with an $E_{V}$ ≥ 40 J/mm^3^ and remarkable tensile properties were achieved when the $E_{V}$ ranged from 40 J/mm^3^ to 100 J/mm^3^. Additionally, remelting strategies played an important role in the tensile properties of the samples. By improving the interface quality, the remelting strategies mitigated the formation of macroscopic cracks, thereby endowing the samples with excellent mechanical properties, especially for those with an $E_{V}$ ≤ 40 J/mm^3^.

The fracture locations of the tensile specimens were recorded and are shown in Figure 5. Only the sample fabricated using the remelting strategy with a volumetric energy density of 180.47 J/mm^3^ fractured at the CoCrMo end, which could be attributed to thermal stress-induced cracks. An excessive laser energy input may generate large thermal stress and form keyhole pores, deteriorating the mechanical properties. The volumetric energy density also played a crucial role in the interface properties. Interface fractures occurred in samples with a low $E_{V}$ (≤ 50 J/mm^3^), indicating that the energy density was required to exceed 50 J/mm^3^ to ensure the formation of strong and robust metallurgical bonding between CoCrMo and CuZrCr.

### 3.2. Hardness Test

Hardness tests were performed at the CuZrCr end, CoCrMo end, and at the interface. The results are listed in Table 3. The hardness value was recorded as 0 for samples that failed before testing.

Variations in the hardness values from different volumetric energy densities were analyzed and are shown in Figure 6.

Figure 6 shows that the hardness of the CuZrCr end exhibited slight variations with changes to the $E_{V}$ and laser scanning strategy. At the interface, the hardness slightly increased with an increase in the $E_{V}$. As the energy density rose, the laser energy input increased, leading to a more thorough mixing of the elements at the interface. This potentially facilitated the formation of a higher content of hard and brittle intermetallic compounds, resulting in an increased interface hardness. Conversely, the hardness of the CoCrMo end slightly decreased with an increase in the energy density. The increased laser energy input led to an increase in the size of the melt pool and the temperature of the molten metal, which may have caused grain coarsening during the cooling process, consequently resulting in a decreased hardness.

### 3.3. Interface Characterization

An optical microscope (OM) was used to characterize the interface of the multi-metal parts. As shown in Figure 7, for samples without laser remelting, microcracks developed near the interface region when the energy density was above 43 J/mm^3^, which could be attributed to the thermal stress induced during rapid cooling. These cracks typically propagated along the building direction, explicable by the temperature gradient that occurred during the L-PBF process. The CuZrCr substrate, characterized by high thermal conductivity, resulted in a temperature gradient that closely aligned with the building direction. The non-uniform thermal expansion induced by the temperature gradient generated localized residual stress within the deposited CoCrMo, leading to the emergence of cracks. When the energy density decreased below 43 J/mm^3^, the number of cracks significantly decreased and pores were observed, indicating a lack of fusion during the L-PBF process. When the energy density was low, the liquid metal may have had poor flowability or there was insufficient liquid metal to completely backfill the volume shrinkage during the solidification process, resulting in pores and cracks. When the energy density further decreased to 20 J/mm^3^, continuous cracks propagated along the interface between the CuZrCr end and CoCrMo end, which was attributed to incomplete fusion on the interface during the initial few layers of deposition. As Cu-based alloys have high thermal conductivity and reflectivity to infrared light, a low laser energy input resulted in poor metallurgical bonding at the interfacial region, making it more susceptible to thermal stress during the subsequently cooling, which corresponded well with the results obtained from the tensile tests.

In the samples subjected to laser remelting within the initial three deposition layers, notable disparities were discernible in the OM image. By enhancing the thermal energy input during the manufacturing process, the laser remelting strategy could increase the fusion zone of the CuZrCr substrate and facilitate a more thorough diffusion between the CuZrCr and CoCrMo alloys. For samples fabricated with a volumetric energy density of 108 J/mm^3^, the depth of the fusion zone increased from 81 μm to 107 μm. Additionally, both the density number and size of defects near the interface were significantly reduced. Lastly, laser remelting markedly improved the interfacial properties of samples, especially those fabricated at low volumetric energy densities. By increasing the laser energy input, thereby increasing the amount of remelted copper, more robust metallurgical bonding occurred at the interface, consequently leading to crack propagation along the interface.

## 4. Conclusions

In this study, optimized process parameters to deposit CoCrMo onto a CuZrCr substrate via L-PBF and the influence of laser remelting were investigated. A total of 27 printing parameters and 2 strategies were compared. Detailed analyses on the relationship between the $E_{V}$ and mechanical properties, including tensile properties and hardness, were conducted.

The results showed that the process parameters and laser scanning strategies had a significant influence on the mechanical properties and interface characteristics. Samples fabricated with an $E_{V}$ ≤ 20 J/mm^3^ had large pores and interfacial cracks attributed to a lack of fusion, thereby showing slight tensile ductility. As the $E_{V}$ increased to a range between 40 J/mm^3^ and 100 J/mm^3^, the samples obtained the desired mechanical properties, with a strong interface combining the alloys. However, an excessive $E_{V}$ resulted in cracks due to the large thermal stress, deteriorating the strength. A comparison between samples fabricated using different laser scanning strategies indicated that laser remelting significantly improved the interface properties, especially when the $E_{V}$ was below 40 J/mm^3^, endowing samples with better mechanical properties. Hardness tests showed that with an increase in $E_{V}$, there was no significant change in the hardness at the CuZrCr end. However, the hardness at the interface and the CoCrMo end showed an increasing and decreasing trend with an increase in $E_{V}$, respectively. Interface characterization was performed using an optical microscope. The results showed that when the $E_{V}$ was above 43 J/mm^3^, the main defects in the L-PBF CoCrMo samples were thermal cracks, which gradually changed to pores with a lack of fusion when the $E_{V}$ decreased. Cracks propagated along the interface of samples fabricated using an $E_{V}$ ≤ 20 J/mm^3^ due to weak interface bonding. The observations and conclusions summarized in this study can effectively be used to fabricate CuZrCr–CoCrMo dissimilar alloy parts.

## Figures and Tables

**Figure 1 micromachines-15-00765-f001:**
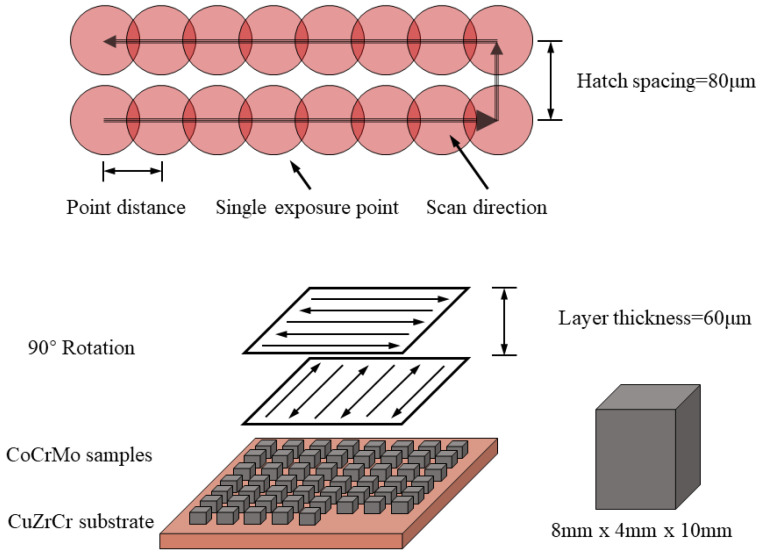
Working principle of pulsed wave laser.

**Figure 2 micromachines-15-00765-f002:**
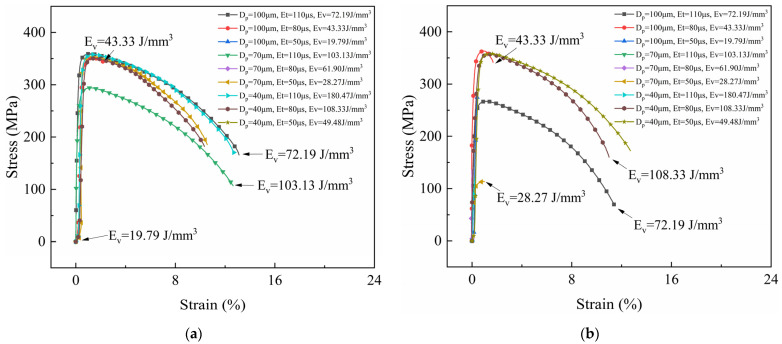
Stress–strain curves of CuZrCr/CoCrMo samples with laser power of 350 W. (**a**) No remelting; (**b**) laser remelting of the first three layers of powder thrice.

**Figure 3 micromachines-15-00765-f003:**
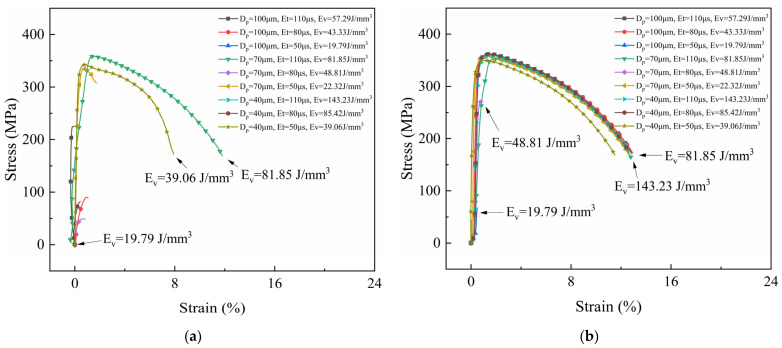
Stress–strain curves of CuZrCr/CoCrMo samples with laser power of 275 W. (**a**) No remelting; (**b**) laser remelting of the first three layers of powder thrice.

**Figure 4 micromachines-15-00765-f004:**
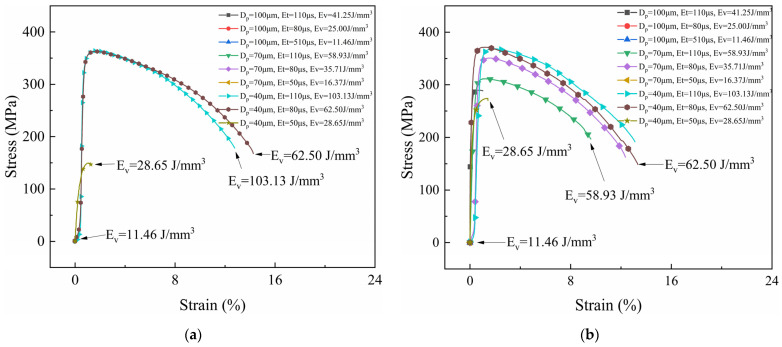
Stress–strain curves of CuZrCr/CoCrMo samples with laser power of 200 W. (**a**) No remelting; (**b**) laser remelting of the first three layers of powder thrice.

**Figure 5 micromachines-15-00765-f005:**
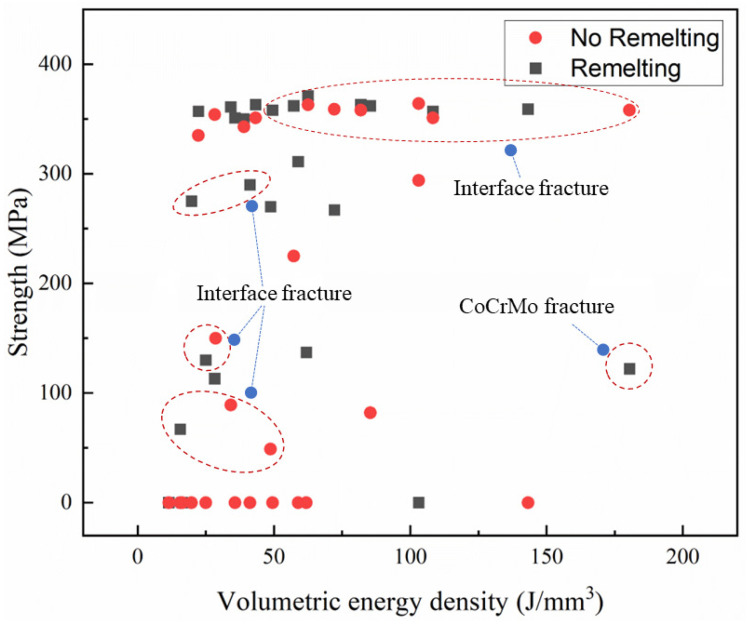
Variations in tensile strength of CuZrCr/CoCrMo samples with different volumetric energy densities (*E*_v_).

**Figure 6 micromachines-15-00765-f006:**
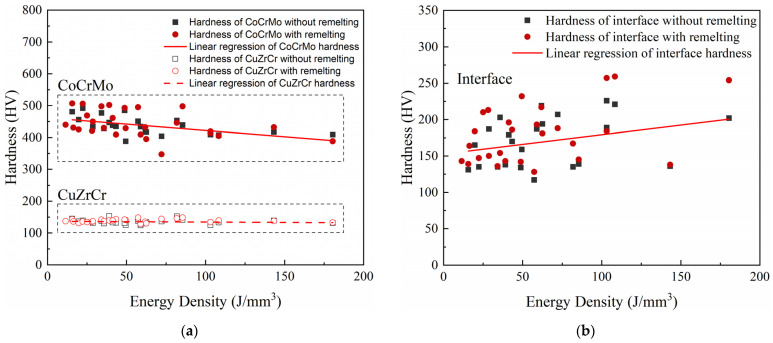
Variations in hardness at (**a**) CuZrCr end and CoCrMo end and (**b**) interface of CuZrCr/CoCrMo samples with different volumetric energy densities (*E*_v_).

**Figure 7 micromachines-15-00765-f007:**
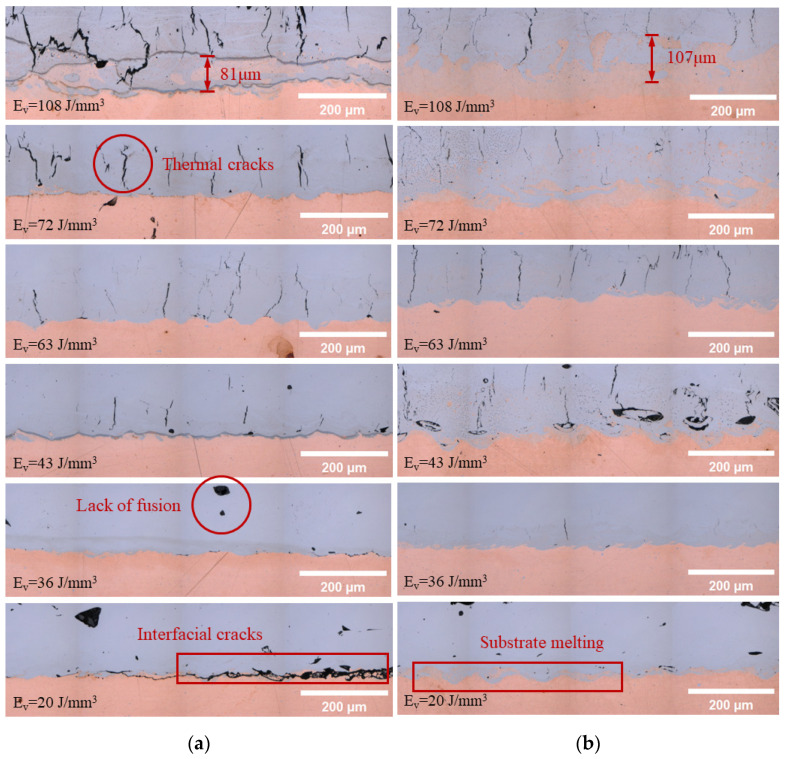
Optical microscope images of the interfacial region of samples with different volumetric energy densities without laser remelting (**a**) and with laser remelting (**b**).

**Table 1 micromachines-15-00765-t001:** L-PBF process parameters and volumetric energy density of all samples.

Process Parameter No.	P (W)	$D_{p}$ (μm)	$E_{t}$ (μs)	V (mm/s)	$E_{v}$ (J/mm^3^)
1	350	100	110	910	72.19
2			80	1250	43.33
3			50	2000	19.79
4		70	110	640	103.13
5			80	880	61.90
6			50	1400	28.27
7		40	110	360	180.47
8			80	500	108.33
9			50	800	49.48
10	275	100	110	910	57.29
11			80	1250	34.17
12			50	2000	15.63
13		70	110	640	81.85
14			80	880	48.81
15			50	1400	22.32
16		40	110	360	143.23
17			80	500	85.42
18			50	800	39.06
19	200	100	110	910	41.25
20			80	1250	25.00
21			50	2000	11.46
22		70	110	640	58.93
23			80	880	35.71
24			50	1400	16.37
25		40	110	360	103.13
26			80	500	62.50
27			50	800	28.65

**Table 2 micromachines-15-00765-t002:** Tensile properties of all samples.

Process Parameter No.	Strength (MPa)		Strain (%)		Fracture Location	
	Remelting	No Remelting	Remelting	No Remelting	Remelting	No Remelting
1	267	359	11.39	13.12	CuZrCr	CuZrCr
2	363	351	1.68	2.21	Interface	Interface
3	275	0	0.41	0	Interface	Failure
4	0	294	0	12.64	Failure	CuZrCr
5	137	0	0.74	0	CuZrCr	Failure
6	113	354	1.00	10.57	CuZrCr	CuZrCr
7	122	358	0.30	12.72	Interface	CuZrCr
8	357	351	10.99	10.33	CuZrCr	CuZrCr
9	358	0	12.70	0	CuZrCr	Failure
10	362	225	12.84	0.64	CuZrCr	CuZrCr
11	361	89	12.94	1.05	CuZrCr	Interface
12	67	0	0.50	0	Interface	Failure
13	363	358	12.79	11.78	CuZrCr	CuZrCr
14	270	49	0.76	0.81	CuZrCr	Interface
15	357	335	12.56	1.71	CuZrCr	Interface
16	359	0	12.70	0	CuZrCr	Failure
17	362	82	12.58	0.44	CuZrCr	CoCrMo
18	350	343	11.50	7.91	CuZrCr	CuZrCr
19	290	0	1.02	0	Interface	Failure
20	130	0	0.14	0	Interface	Failure
21	0	0	0	0	Failure	Failure
22	311	0	9.60	0	CuZrCr	Failure
23	351	0	12.30	0	CuZrCr	Failure
24	0	0	0	0	Failure	Failure
25	368	363	13.18	12.77	CuZrCr	CuZrCr
26	371	363	13.34	14.26	CuZrCr	CuZrCr
27	274	150	1.42	0.81	Interface	Interface
Average	257	163.9	6.64	4.21	/	/
Standard Deviation	131.2	164.3	5.98	5.59	/	/

**Table 3 micromachines-15-00765-t003:** Hardness of all samples.

Hardness (HV)	Interface		CoCrMo		CuZrCr	
Process Parameter No.	Remelting	No Remelting	Remelting	No Remelting	Remelting	No Remelting
1	188	207	347	404	144	136
2	186	170	409	435	143	132
3	184	165	425	456	131	136
4	257	226	419	410	134	132
5	217	219	432	417	135	130
6	213	0	421	0	135	0
7	254	202	388	409	133	131
8	259	221	405	407	139	133
9	232	159	429	388	141	125
10	128	117	495	451	148	138
11	136	135	498	477	142	137
12	139	131	507	481	143	145
13	167	135	446	453	146	152
14	142	134	493	485	142	131
15	147	135	506	492	136	139
16	138	136	432	417	138	139
17	145	139	498	439	148	141
18	143	138	502	447	139	153
19	196	179	461	438	139	133
20	210	0	469	0	134	0
21	143	0	440	0	137	0
22	193	187	409	434	128	124
23	154	203	430	428	137	130
24	164	0	431	0	136	0
25	164	189	413	409	134	125
26	181	194	395	417	130	134
27	150	187	450	434	137	131

## Data Availability

The original contributions presented in the study are included in the article, further inquiries can be directed to the corresponding authors.

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
