# Peer review of "Mechanical Properties and Interfacial Characterization of Additive-Manufactured CuZrCr/CoCrMo Multi-Metals Fabricated by Powder Bed Fusion Using Pulsed Wave Laser"

_micromachines, 2024, doi:10.3390/mi15060765_

Round 1

Reviewer 1 Report

Comments and Suggestions for Authors

This manuscript describes a typical process optimization experiment on additive manufacturing of CuZrCr/CoCrMo by powder bed fusion using pulsed wave laser. The authors claim that by tuning a combination of 5 factors, like power, scam velocity, they strived to obtain the optimum mechanical properties at Ev 43 J/mm3. They observed that the main defect was thermal cracks. Overall, I believe this work warrants reading for the additive manufacturing community. However, I do have concerns shown below:

1. This work is clearly a typical process optimization experiment, which is very common in industry to control the product quality. Considering this context, the ways the authors design the experiment and present the data are rather naïve. I am shocked that throughout the whole paper, no statistical analysis is mentioned.

2. Following last comment, at page 4, table 1, there are five factors, i.e. variables, and at page 5, table 2, three responses even with two level of changes. Even for an experienced industrial engineer, to design such an experiment to get convincing results is a daunting task. However, I can tell that the authors have no knowledge of design of experiment, in statistical sense, whatsoever. Simply presenting data like Figure 2 to 4 means nothing statistical significant without proper statistical analysis.

3. Even for a designed experiment, we have to analyze the data with the aid of professional statistical software, like SAS, JMP, Minitab, R, to name a few. However, I don’t see any of these tools being used in this work, which makes the conclusion basically meaningless.

4. I do appreciate the efforts the authors put in this process optimization. I sincerely suggest the authors consult an industry professional, especially quality control engineer, for further insight of their data.

Author Response

Point-by-point response to Comments and Suggestions for Authors

Comments 1: This work is clearly a typical process optimization experiment, which is very common in industry to control the product quality. Considering this context, the ways the authors design the experiment and present the data are rather naïve. I am shocked that throughout the whole paper, no statistical analysis is mentioned.

Response 1: Thank you for pointing this out. We acknowledge your concern regarding the experimental design and the absence of statistical analysis in our initial submission. We have now incorporated the descriptive statistical analysis using Minitab and the regression analysis using origin. The revised manuscript now includes these statistical analyses to strengthen the validity of our findings.

Comments 2: Following last comment, at page 4, table 1, there are five factors, i.e. variables, and at page 5, table 2, three responses even with two level of changes. Even for an experienced industrial engineer, to design such an experiment to get convincing results is a daunting task. However, I can tell that the authors have no knowledge of design of experiment, in statistical sense, whatsoever. Simply presenting data like Figure 2 to 4 means nothing statistical significant without proper statistical analysis.

Response 2: Thank you for your feedback. We agreed that simply presenting data without statistical analysis in page 5, table 2 lack persuasiveness. We have added descriptive statistical analysis including average value and standard deviation value at the end of page 5, table 2. But the experimental design on process parameters listed in table 4, page 1 only has three independent variables: laser power, point distance, and exposure time. The velocity and energy density are calculated using these variables. The influence of remelting strategy is investigated in this study. And the “Remelting”/”No remelting” column in table 5, page 2 do not represent responses but different strategies with the same process parameters.

Comments 3: Even for a designed experiment, we have to analyze the data with the aid of professional statistical software, like SAS, JMP, Minitab, R, to name a few. However, I don’t see any of these tools being used in this work, which makes the conclusion basically meaningless.

Response 3: Thanks for your comments on the manuscript, we have taken your suggestion and used Minitab to conduct descriptive statistical analysis in page 5, table 2, and used origin to do the linear regression in page 9, figure 6 to better support our conclusion.

Reviewer 2 Report

Comments and Suggestions for Authors

The article is primarily a parameter study on the additive manufacturing of multi-materials. The topic is highly interesting and will become even more relevant in the future but the choice of the two selected materials and their potential fields of application is not well presented.

In addition, readability is hindered by the fact that many, but by no means all, abbreviations are explained before first use, spaces are often missing (especially before the citation []), and the tenses are mixed up. For example, the past tense is often used instead of the present tense. Furthermore, especially on pages 4 and 5, all references to figures are incorrect and the diagrams are much too small, sometimes blurred, and difficult to read. It also remains unclear how many samples were tested in the tensile test, what the deviation of the samples/results is, and what forces were used in the hardness test. It would be interesting to see a fracture surface of the tensile specimens to better understand the effects in the diagram.

 A question is why the usual rotation of 60° was deviated from and 90° was chosen.

In Figure 7, the last image shows “element interdiffusion” - this is not visible in the illustrated section. More in-depth investigations (SEM?) would be useful here and should be added. This would also show the extent to which the comparison between the two states affects the joining zone.

There are also a few editorial issues:

It is uncommon to use the formulation “we”. There should be a space between the number and the unit. Likewise, care should be taken to use standardized superscripts and subscripts for indices, as this gets completely mixed up in the especially in the abstract.

I strongly recommend a linguistic revision and a minor revision of the experimental part.

Comments on the Quality of English Language

I strongly recommend a linguistic revision and a minor revision of the experimental part.

Author Response

Point-by-point response to Comments and Suggestions for Authors

(can also be seen in the attachment)

Comments 1: The choice of the two selected materials and their potential fields of application is not well presented.

Response 1: Thank you for pointing this out. The introduction part have been expanded to fully present the reasons behind our material choices and their potential applications.

Revised in page 1, paragraph 2

Combining copper alloy and cobalt alloy can achieve a synergistic enhancement of material properties, leveraging the excellent thermal conductivity of copper alloys and the high-temperature mechanical properties and corrosion resistance of cobalt alloys. This unique combination of properties makes the dissimilar alloy highly suitable for a wide range of advanced applications. For examples, in the aerospace industry, this dissimilar alloy can be used in the construction of engine components and other critical parts that must withstand extreme temperatures and corrosive environments. In the nuclear industry, the superior thermal conductivity and corrosion resistance are crucial for the reliability and safety of nuclear reactors. Additionally, Cu/Co multi alloy parts can be used in thermal management systems, where both high thermal conductivity and mechanical robustness are essential

Comments 2: Abbreviations are explained before first use, spaces are often missing (especially before the citation []), and the tenses are mixed up.

Response 2: A thorough review of the manuscript have been conducted to ensure that all abbreviations are defined upon first use, added missing spaces, particularly before citations, and standardized the use of tenses. We have consistently used the present tense to describe current research findings and conclusions.

Comments 3: Especially on pages 4 and 5, all references to figures are incorrect and the diagrams are much too small, sometimes blurred, and difficult to read.

Response 3: All figure references, especially those on pages 4 and 5, have been corrected. Additionally, we have redesigned the figures to be clearer, ensuring that they are easy to read and interpret.

Comments 4: It also remains unclear how many samples were tested in the tensile test, what the deviation of the samples/results is, and what forces were used in the hardness test. It would be interesting to see a fracture surface of the tensile specimens to better understand the effects in the diagram.

Response 4: A more detailed information regarding the number of samples tested in the tensile tests and the forces used in the hardness tests are provided. However, the tensile tests were tested only once and the characterization of fracture morphology still remains to be investigated.

Mechanical tests were carried on a total of 54 samples fabricated using 27 combinations of different process parameters and two scanning strategies. in page 4, paragraph 3, line 1.

“FutureTech FM-300e Microhardness Tester were used to analyze the microhardness of the multi-metal samples with a specific load of 300gf.” in page 5, paragraph 2, line 3.

Comments 5: Why the usual rotation of 60° was deviated from and 90° was chosen.

Response 5: According to Zhou’s work “Effect of scanning strategies on the microstructure and mechanical properties of Ti–15Mo alloy fabricated by selective laser melting”, increasing the rotation angle between layers could induce the formation of fine grains, so we choose 90°instead of 60°or 67°.

Comments 6: In Figure 7, the last image shows “element interdiffusion” - this is not visible in the illustrated section. More in-depth investigations (SEM?) would be useful here and should be added. This would also show the extent to which the comparison between the two states affects the joining zone.

Response 6: Thanks for pointing this out. The results of metallurgical microscopy represents a more macroscopic phenomenon. While element interdiffusion belongs to microscopic category, which should be verified through in-depth characterization. Accordingly, We have, modified the term “element interdiffusion” to “substrate melting” in Figure 7.

Comments 7: It is uncommon to use the formulation “we”. There should be a space between the number and the unit. Likewise, care should be taken to use standardized superscripts and subscripts for indices, as this gets completely mixed up in the especially in the abstract.

Response 7: A thorough review of the manuscript have been conducted to avoid using informal expressions such as "we" and ensured there is a space between numbers and units. And the use of superscripts and subscripts throughout the manuscript are standardized.

Round 2

Reviewer 1 Report

Comments and Suggestions for Authors

All my concerns have been addressed.